# Comparative Analysis of miRNA-mRNA Regulation in the Testes of *Gobiocypris rarus* following 17α-Methyltestosterone Exposure

**DOI:** 10.3390/ijms24044239

**Published:** 2023-02-20

**Authors:** Shaozhen Liu, Junliang Zhou, Qiong Yang, Yue Chen, Qing Liu, Weiwei Wang, Jing Song, Xianzong Wang, Yu Liu

**Affiliations:** College of Animal Science, Shanxi Agricultural University, Jinzhong 030800, China

**Keywords:** 17α-methyltestosterone, *Gobiocypris rarus*, RNA-Seq, miRNA-seq, testis

## Abstract

17α-Methyltestosterone (17MT), a synthetic organic compound commonly found in sewage waters, can affect reproduction in aquatic animals, such as tilapia and yellow catfish. In the present study, male *Gobiocypris rarus* were exposed to 25, 50, and 100 ng/L of 17α-methyltestosterone (17MT) for 7 days. We first analyzed miRNA- and RNA-seq results to determine miRNA-target gene pairs and then developed miRNA-mRNA interactive networks after 17MT administration. Total weights, total lengths, and body lengths were not significantly different between the test groups and control groups. The paraffin slice method was applied to testes of *G. rarus* in the MT exposure and control groups. We found that there were more mature sperm (S) and fewer secondary spermatocytes (SSs) and spermatogonia (SGs) in the testes of control groups. As 17MT concentration increased, fewer and fewer mature sperm (S) were observed in the testes of male *G. rarus*. The results showed that FSH, 11-KT, and E2 were significantly higher in individuals exposed to 25 ng/L 17MT compared with the control groups. VTG, FSH, LH, 11-KT, and E2 were significantly lower in the 50 ng/L 17MT exposure groups compared to the control groups. VTG, FSH, LH, 11-KT, E2, and T were significantly lower in the groups exposed to 100 ng/L 17MT. High-throughput sequencing revealed 73,449 unigenes, 1205 known mature miRNAs, and 939 novel miRNAs in the gonads of *G. rarus*. With miRNA-seq, 49 (MT25-M vs. Con-M), 66 (MT50-M vs. Con-M), and 49 (MT100-M vs. Con-M) DEMs were identified in the treatment groups. Five mature miRNAs (miR-122-x, miR-574-x, miR-430-y, lin-4-x, and miR-7-y), as well as seven differentially expressed genes (*soat2*, *inhbb*, *ihhb*, *gatm*, *faxdc2*, *ebp*, and *cyp1a1*), which may be associated with testicular development, metabolism, apoptosis, and disease response, were assayed using qRT-PCR. Furthermore, miR-122-x (related to lipid metabolism), miR-430-y (embryonic development), lin-4-x (apoptosis), and miR-7-y (disease) were differentially expressed in the testes of 17MT-exposed *G. rarus.* This study highlights the role of miRNA-mRNA pairs in the regulation of testicular development and immune response to disease and will facilitate future studies on the miRNA-RNA-associated regulation of teleost reproduction.

## 1. Introduction

17α-Methyltestosterone (17MT), a synthetic organic compound commonly found in sewage waters from paper mills, domestic sewage, and livestock manure, when released into the environment can affect reproduction in aquatic animals, such as tilapia and yellow catfish. Approximately 4.1–7.0 ng/L 17MT was detected in wastewater samples from Beijing. Furthermore, aromatase transforms 17MT to estrogen-like 17β-estradiol, which exerts estrogenic effects [1]. 17MT exposure can thus lead to sex reversal in many fish species, such as zebrafish and orange-spotted grouper [2,3]. Exposure to 17MT can affect the sex-steroid hormone levels in the body by preventing the activity of steroidogenic enzymes [3]. Previous studies have shown impaired testicular development in rare minnows (*Gobiocypris rarus*) exposed to 17MT (25–100 ng/L) [4,5]. Moreover, 302.5 ng/L 17MT disturbed gene expression in the hypothalamus–pituitary–gonadal axis of mummichog (*Fundulus heteroclitus*) [6].

MicroRNAs (miRNAs), which are small regulatory molecular RNAs 21–23 nucleotides in length [7], play an important role in regulating sex determination and differentiation in the gonads of dark sleepers [8], metabolism in darkbarbel catfish and zebrafish and disease prevention [9,10,11]. Five biased miRNAs (let-7a, miR-10a, miR-20a, miR-130a, and miR-202) related to egg quality in Atlantic salmon (*Salmo salar*) and miRNAs related to gonadal development (zebrafish miR-430; *Trachinotus ovatus* miR-143, 101a, 202-5p, 181a-5p, and let-7c-5p; common carp miR-24, 146a, 192, 21, 143, and 454b; *Epinephelus coioides* miRNA-26a; and blunt snout bream let-7a/b/d) have recently been discovered through miRNA-RNA crosstalk [12,13,14,15,16,17]. However, studies focusing on the testes, such as those on Atlantic salmon, are scarce, and most studies have only focused on gene expression, without considering the underlying miRNA regulation [18]. Studies involving miRNA-mRNA analysis in Japanese flounder have shown that steroid-hormone-synthesis-related pathways play an important role in such reproduction processes [19].

Herein, we first analyzed miRNA- and RNA-seq results to determine miRNA-target gene pairs and then developed miRNA-mRNA interactive networks after 17MT administration to reveal potential regulatory mechanisms of reproduction and immune response to disease in rare minnows.

## 2. Results

### 2.1. Morphological Changes

The measured total weights, total lengths, and body lengths (*n* = 6) are listed in Table 1. The total weights, total lengths, and body lengths were not significantly different between the test groups and control groups (*p* > 0.05). We found there were more mature sperm (S) and fewer secondary spermatocytes (SSs) and spermatogonia (SGs) in the testes of the control groups (Figure 1a). As 17MT concentration increased, fewer and fewer mature sperm (S) were observed in the testes of male *G. rarus* (Figure 1b–d). The results showed that with increasing secondary spermatocytes (SSs) and spermatogonia (SGs) in the testes of the 17MT exposure groups, mature sperm count decreased in comparison to the control groups (Figure 1b–d).

### 2.2. Sex-Steroid Hormone Activity

Sex-steroid hormone activity levels (*n* = 3) of male *G. rarus* in response to 17MT are shown in Table 1. The standard curves for Vtg, FSH, 11-KT, E2, LH, and T yielded R^2^ values of 0.981, 0.999, 0.995, 0.997, 0.995, and 0.995, respectively. FSH (*p* < 0.05), 11-KT (*p* < 0.01), and E2 levels (*p* < 0.01) in male *G. rarus* were significantly higher in individuals exposed to 25 ng/L 17 MT compared to the control groups. Vtg (*p* < 0.05), FSH (*p* < 0.01), LH (*p* < 0.01), 11-KT (*p* < 0.01), and E2 levels (*p* < 0.01) of male fish were significantly lower in the groups exposed to 50 ng/L 17MT compared with the control groups. Vtg (*p* < 0.01), FSH (*p* < 0.05), LH (*p* < 0.01), 11-KT (*p* < 0.01), E2 (*p* < 0.05), and T (*p* < 0.05) were significantly lower in the groups exposed to 100 ng/L 17MT compared with the control groups.

### 2.3. RNA- and miRNA-Seq Analyses

In the RNA-seq analysis, gene numbers and ratios were not found to be significantly different between the test groups (*n* = 3) (Appendix A). In the miRNA-seq analysis, total non-coding RNA reads, rRNAs, snRNAs, snoRNAs, tRNAs, known miRNA numbers, novel miRNA numbers, miRNA numbers, and target gene numbers showed no significant differences across the test groups (*n* = 3). Significant DEGs for normalized gene expression among the different 17MT groups were identified. Totals of 59 (MT25-M vs. Con-M), 77 (MT50-M vs. Con-M), and 66 (MT100-M vs. Con-M) genes were identified as significant DEGs in the treatment groups (Appendix A). The MT50-M and MT100-M groups presented 300 and 246 significant DEGs, respectively, in comparison to the MT25-M groups, whereas the MT100-M groups presented 135 significant DEGs when compared to the MT50-M groups. For miRNA-seq, 49 (MT25-M vs. Con-M), 66 (MT50-M vs. Con-M), and 49 (MT100-M vs. Con-M) DEMs were identified in the treatment groups (Appendix A). Of these, 11, 21, and 20 DEMs could be annotated as known miRNAs, whereas 38, 28, and 46 DEMs were annotated as novel miRNAs (Appendix A).

In the RNA-seq analysis, several KEGG pathways, including neuroactive ligand–receptor interactions, cytokine–cytokine receptor interactions, phagosomes, and cell adhesion molecules (CAMs), were found to be enriched (Figure 2b). The five most enriched KEGG pathways were related to metabolism (global and overview, lipid, and amino acid metabolism being the top three) and organismal systems (endocrine, circulatory, and immune systems) in both the RNA- and miRNA-seq analyses. In the RNA-seq analysis, 17MT concentration explained the largest fraction of the variation (22.0% along PC1, *p* < 0.05; Appendix A) after accounting for the variation present. Approximately 13.6% of the variation was explained by PC2, and 9.4% of the variation was explained by PC3. Similarly, for miRNA, 17MT concentration explained the largest fraction of the variation (36.5% along PC1, *p* < 0.05; Appendix A), after accounting for the variation present, and 14.8% and 13.0% of the variation was explained by PC2 and PC3, respectively.

### 2.4. STEM Analysis

In the RNA-seq analysis, 565, 896, 1983, 473, 402, 276, 935, 973, 823, 394, and 173 DEGs were identified in profiles 9, 10, 11, 13, 15, 16, 17, 18, 19, 22, and 25, respectively, by STEM analysis (*p* < 0.05; Appendix A and Figure 2d). All these DEGs were enriched in several immune pathways (Appendix A), including IgA production of the intestinal immune network and the Toll-like receptor signaling pathway (profile 11 in particular), and disease-related pathways, such as the Jak-STAT signaling pathway, cytokine and cytokine receptor interactions, phagosomes, and cell adhesion molecules (CAMs) (profile 11). Biosynthesis pathways of amino acids, antibiotics, secondary metabolites, and carbon were enriched in profile 15. Steroid biosynthesis and retinol and drug (through cytochrome P450) metabolism were also enriched in profile 15. Metabolic pathways (antibiotics and secondary metabolites), regulation of actin cytoskeleton, neuroactive ligand–receptor interactions, phagosomes, and cell adhesion molecules (CAMs) were enriched in almost all profiles (Figure 2c). In the miRNA-seq analysis, 51, 12, 17, and 14 DEMs were identified in profiles 5, 11, 12, and 13, respectively (*p* < 0.05; Appendix A and Figure 2b).

Nonmonotonic dose–response curves were visualized to express the nonlinear relationships between doses and effects [20], and pathways at different times and concentrations of 17MT exposure were evaluated by STEM analysis. Genes involved in several pathways (Figure 3), such as RNA transport, metabolic pathways, cardiac muscle contraction, mineral absorption, and oxidative phosphorylation in particular, were significantly induced on exposure to 17MT when compared to the controls (C1~C3). IgA production of the intestinal immune network (only in MT25-M), cytokine–cytokine receptor interactions and the related chemokine signaling pathways (only in MT50-M), and primary bile acid biosynthesis (only in MT50-M) were identified as biomarkers for the respective groups. In the MT100-M groups with the highest 17MT concentrations, the TGFβ signaling pathway, proteasomes, arginine and proline metabolism, and biosynthesis pathways of arginine, amino acids, antibiotics, and secondary metabolites were identified as biomarkers. After exposure to increasing concentrations of 17MT (C4~C6), almost the same pathways were enriched in the MT50/100-M groups, with only steroid biosynthesis being enriched in C6, implying that exposure to 50~100 ng/L of 17MT can alter testicular steroid hormones, levels of which can be employed as biomarkers for early detection. Phagosomes, the Ras signaling pathway, proteasomes, and biosynthesis pathways of antibiotics, histidine, phenylalanine, purine, tryptophan and metabolism, IgA production of the intestinal immune network, the PPAR signaling pathway, and regulation of lipolysis in adipocytes were enriched in the MT100-M groups when compared to the MT50-M groups. Pathways of apoptosis, Jak-STAT, hedgehog, PI3K-Akt and Toll-like receptors, microbial metabolism in diverse environments, pyruvate metabolism, and salivary secretion were enriched in the comparison between MT50-M and MT25-M (Figure 3).

Regarding these miRNAs and RNAs, profiles 5 and 12 (*p* < 0.001), profile 13 (*p* < 0.01), and profile 11 (*p* < 0.05) of the miRNAs showed the strongest associations with profile 15 (*p* < 0.001) of the RNAs. The miRNA-target gene interaction network constructed using WGCNA showed the highest occurrence in the black (disease-related), blue (metabolism), and turquoise (oxidative and protein ubiquitin) pathways (Appendix A). In the black group, apoptosis, IgA production of the intestinal immune network, cytokine and cytokine receptor interactions, regulation of actin cytoskeleton, phagosomes, and cell adhesion molecules (CAMs) were enriched. In the blue group, metabolic pathways, ubiquitin-mediated proteolysis, and biosynthesis pathways of antibiotics and secondary metabolites and secondary metabolites and other pathways listed in Appendix A were enriched, whereas in the turquoise group, metabolic pathways, oxidative phosphorylation, and ubiquitin-mediated proteolysis were enriched.

A total of 24,326 miRNA-mRNA pairs with negative correlations were identified. Moreover, via RNA-seq, in profile 15, 56 mRNA-miRNA pairs were identified, and a further 10 miRNA-mRNA pairs were identified in profiles 5 and 12 via miRNA-seq. Therefore, when miRNAs are induced by 17MT, their target mRNAs are downregulated and vice versa.

### 2.5. qRT-PCR Validation of DEGs and DEMs

Finally, we selected seven negative miRNA-mRNA interactions with six mature miRNAs (miR-122-x, miR-574-x, miR-430-y, miR-217-x, lin-4-x, and miR-7-y) and seven validated mRNAs (*soat2*, *inhbb*, *ihhb*, *gatm*, *faxdc2*, *ebp*, and *cyp1a1*). The sequencing results were consistent with those of the qRT-PCR validations (*n* = 3) (Appendix A). Several genes played critical roles in multiple pathways. For example, miRNA targets of the *faxdc2* gene were miR-122-x and miR-574-x, and miRNA targets of *inhbb* were miR-430-y, lin-4-x, and miR-7-y, which were involved in TGFβ signaling (Figure 4). In addition, *soat2*, *inhbb*, *ebp*, *faxdc2*, *cyp1a1*, miR-574-x, miR-430-y, miR-122-x, and miR-217-x were downregulated in the MT25-M groups; miR-574-x and miR-7-y were downregulated in the MT50-M and MT100-M groups; and miR-430-y, miR-217-x, and lin-4-x were downregulated in the MT100-M groups (Figure 5).

lin-4-x and miR-7-y were found to have negative correlations with *faxdc2*, *inhbb*, *ihhb*, *soat2*, and gatm, and miR-574-x was found to have a significant negative correlation with *faxdc2* (Appendix A).

### 2.6. Targeting Relationship Validations

We found that the 3′-UTR of *faxdc2* contains potential miR122 and miR-574-x binding sites, that the 3′-UTR of *inhbb* contains potential miR-430-y, lin-4x, and miR-7-y binding sites, and that the 3′-UTR of *ihhb* contains a potential miR-217x binding site (Appendix A).

When miR122 and miR-574-x were co-transfected with faxdc2-WT, when miR-430-y, and lin-4x and miR-7-y were co-transfected with inhbb-WT, luciferase activity decreased extremely significantly compared to the control groups (Figure 6). When miR122 was co-transfected with faxdc2-MuT, miR-574-x with faxdc2-MuT, miR-430-y with inhbb-MuT, lin-4x with inhbb-MuT, and miR-7-y with inhbb-MuT, there were no statistical differences compared with the control group. These results showed that miR-122 and miR-574-x target faxdc2 and that miR-430-y, lin-4x, and miR-7-y target inhbb, which is consistent with our bioinformatics analysis of miRNA target gene predictions. However, when miR-217-x was co-transfected with ihhb-WT, luciferase activity did not decrease, indicating that ihhb is not a target gene of miR-217-x (Figure 6).

## 3. Discussion

Several studies have shown that testicular development is suppressed with 17MT exposure in Atlantic salmon [18]; however, the involvement of the miRNA-mRNA regulatory network in this phenomenon has not yet been reported. In the present study, we found that hormone levels, transcripts involved in metabolism/apoptosis/disease response, particularly gene expression in oxidative and protein ubiquitin (proteasome) pathways, steroid biosynthesis in testicular development, and biosynthesis pathways of arginine, amino acids, antibiotics, and secondary metabolites, were significantly affected by the highest administered dose of 17MT. This was further supported by integrated miRNA-mRNA analysis using STEM.

Seven genes (*soat2*, *inhbb*, *ihhb*, *gatm*, *faxdc2*, *ebp*, and *cyp1a1*) and their targets and mature miRNAs (miR-122-x, miR-574-x, miR-430-y, lin-4-x, and miR-7-y) were differentially expressed, which was verified by qRT-PCR. Of these, *soat2*, *inhbb*, *ebp*, *cyp1a1*, miR-574-x, and miR-430-y were downregulated in the MT25-M groups, miR-574-x and miR-7-y were downregulated in the MT50-M and MT100-M groups, and *miR-430-y* and *lin-4-x* were downregulated in the MT100-M groups. It has previously been shown that 17MT can alter steroid biosynthesis and lipid metabolism in mice, fish species such as rainbow trout (*Salmo gairdneri*), and ramshorn snails (*Marisa cornuarietis*) [21,22,23]. The miRNA targets of the *faxdc2* gene were miR-122-x and miR-574-x. faxdc2 may be employed as a potential gene marker to improve feed efficiency [24]. Its encoded protein, Faxdc2, plays an important role in the development of megakaryocytes, and their dysregulation may contribute to abnormal hematopoietic cell development in leukemia [25]. miR-122-x has been reported to play a role in hepatic lipid metabolism and associated tissue damage (microcystin-LR exposure resulting in apoptosis [26,27,28]; through the inflammatory pathway [29,30,31,32,33]. Studies have shown that miR-122 is involved in the regulation of glycolysis and could be a potential biomarker of cholesterol metabolism in rainbow trout [34,35,36]. It also plays an important role in lipid metabolism by targeting *ap-1*/*c-jun* and sterol regulatory element-binding transcription factor 1 (*srebp1*) [37,38]. Recently, miR-122 was found to be involved in osmoregulation in eels [39], disease prevention (*Vibrio anguillarum*) through toll-like receptors, and the RIG-I signaling pathway [40,41]. Moreover, *faxdc2* and miR-217 have been identified as potential carotenoid color candidate genes in *Tropheus duboisi* and *Botia superciliaris* [42,43]. In this study, *faxdc2* and miR-122 were found to be downregulated in the group exposed to the lowest concentration of 17M (MT25-M), while they were upregulated in the groups exposed to higher concentrations of 17MT (MT50/100-M), with the exception of *miR-217*. Thus, different concentrations of 17MT influenced lipid metabolism and disease prevention.

The inhibin beta B chain-like gene (*inhbb*), anti-Müllerian hormone (*amh*), and apolipoprotein E (*apoe*) were suppressed by follicle-stimulating hormone (FSH) in coho salmon and zebrafish [44,45]. Ovary-*gdf9* significantly suppressed the expression of *amh* in somatic cells, whereas it increased the expression of activin beta subunits (*inhbaa* and *inhbb*) in vitro [45]. The miRNA targets of *inhbb* were *miR-430-y*, *lin-4-x*, and *miR-7-y*, which were involved in TGFβ signaling. Previous studies have shown that *miR-430* plays an important role in early embryonic development in zebrafish by targeting Smarca2 or by regulating histone acetylation [46,47,48]. A high proportion of *miR-430* family members participate in maternal RNA clearance during the earliest developmental stage [47,49]. miR-430 presents a potential novel model for over-riding maternal programming under altered environmental conditions [50]. Maternal mRNA cannot degrade in the absence of *miR-430* [51]. miR-430 may target *foxh1* through TGFβ signaling during early embryonic development in zebrafish [13]. lin-4-x also plays an important role in early zebrafish development [52]. *sp1* was found to be a target of miR-7 in fibroblast proliferation during fibroblast-to-myofibroblast transition and may play an important role in zebrafish brain development [53,54]. miR-7 targets *myd88* in host–virus interactions in *Eriocheir sinensis*, and overexpression of miR-7 can inhibit expression of the interleukin enhancer-binding factor 2 homolog (*ilf2*) and interleukin-16-like genes (*il-16l*) [11]. It identified a novel miR-7a target, *YY1*, and demonstrated that novel *miR-7a* regulates zebrafish hepatic lipid metabolism by controlling *YY1* stabilization, regulating the expression of lipogenic signaling pathways, and stimulating the expression of inflammatory genes [55]. Immune host–pathogen interactions in sea cucumber (*Apostichopus japonicus*) and the immune response of red claw crayfish (*Cherax quadricarinatus*) to white spot syndrome virus were shown to involve *miR-7a* [56,57]. In the present study, *inhbb* (MT25-M) and *miR-7-y* (MT50/100-M) were downregulated following 17MT administration, suggesting that 17MT impaired maternal programming, resulting in impaired development, metabolism, and disease prevention.

Deltamethrin was shown to suppress zebrafish hedgehog signaling pathway genes (Indian hedgehog b, *ihhb*) [58]. *ihhb* plays an important role in vertebral column development in fugu (*Takifugu rubripes*) and is also associated with chondrogenic lineage in Atlantic salmon (*Salmo salar* L.) [59,60]. Abnormal ocular morphogenesis was observed in *sox4*-deficient zebrafish, resulting from elevated hedgehog signaling due to increased expression of the hedgehog pathway ligand (*ihhb*) [61]. *ihhb* was found to play a role in testis maturation [18], and target toll-like receptor 1 was found to perceive LPS stimulation and transfer signals to activate the NF-κB pathway or the TLR signaling pathway in miiuy croaker [62]. This suggests that *ihhb* may be vital for hedgehog signaling following 17MT administration. The targeting relationship validation showed that *miR-217-x* did not target *ihhb.* We will study the relationship of *miR-217-x* to other genes following MT exposure in *G. rarus.*

Triglycerides and increased *soat2* transcripts are associated with lipid metabolism and oxidation resistance in Manchurian trout (*Brachymystax lenok*) [63]. *soat2* is expressed in zebrafish testes, and plays a role in cholesteryl ester synthesis, contributing to yolk cholesterol trafficking during zebrafish embryogenesis [64]. 3-beta-hydroxysteroid- Delta(8),Delta(7)-isomerase (*ebp*) catalyzes an intermediate step in the conversion of lanosterol to cholesterol, resulting in drug resistance (cholesterol biosynthesis [65]; Glycine amidinotransferase (*gatm*) synthesis and uptake of creatine may play a role in skeletal muscle metabolism and zebrafish embryogenesis [66,67]. *cyp1a1* catabolizes 17MT in juvenile Atlantic salmon (*Salmo salar*) and in adult *G. rarus* [68,69]. These genes were down- and upregulated in the MT25 and MT50/100-M groups, respectively, indicating that 17MT exposure affected metabolism.

In summary, 73,449 unigenes, 1205 known mature miRNAs, and 939 novel miRNAs of *G. rarus* were identified by integrated mRNA- and miRNA-seq analyses. From these, we successfully identified five miRNA-target pairs, which may possibly be involved in development, metabolic processes, and disease prevention. *faxdc2* was the target gene regulated by miR-122-x and miR-574-x, whereas *inhbb* was the target gene regulated by miR-430-y, lin-4-x, and miR-7-y. In addition, mRNAs and miRNAs involved in early testicular development and innate immune response were tested by qRT-PCR and were identified as novel regulators of metabolism, embryonic development, and disease prevention in *G. rarus*.

## 4. Material and Methods

### 4.1. Ethics Statement

This study was approved by the Shanxi Agricultural University Animal Care and Ethical Committee, China (IACUC no.: SXAU-EAW-2022F.BN.001012001). During the experiment, the fish were humanely treated. Before being sacrificed, all fish were anesthetized using tricaine methanesulfonate (MS222), and every effort was made to minimize suffering.

### 4.2. Experimental Animals

As 17MT is insoluble in water and soluble only in organic solvents, the 17MT stock solution was prepared in anhydrous ethanol. The *G. rarus* used in the experiments were obtained from the same family through artificial fertilization, and males were segregated after sexual maturity. The sexes were distinguished based on the distance between the hind fin and the tail fin, which is smaller in males than in females. If sex could not be determined from the external appearance, the testes of fish were observed following the different 17MT exposure experiments. After grouping, six-month-old *G. rarus* males were domesticated for one week before the experiments. They were then exposed to different concentrations of 17MT (25, 50, and 100 ng/L, corresponding to groups MT25-M, MT50-M, and MT100-M, respectively; treatment groups) or to 0.0001% anhydrous ethanol (control group; Con-M) for 7 days. The concentration and duration of exposure were selected based on previous reports [4,5,6]. All experiments were performed in triplicate. A total of 12 aquariums (80 L volumes) were used, with 25–30 *G. rarus* males in each aquarium, ensuring a ratio of 1 g of fish for every liter of water. Animals were fed regularly with a fixed quantity of feed every day. The semi-static water-exposure method was employed to change half of the water in the aquarium, sucking out the sewage (residual bait and feces) and simultaneously adding the same amount of water with the required amount of 17MT solution to ensure that the 17MT concentration in the aquarium remained unchanged.

### 4.3. Measurement and Sampling of Biological Indicators

#### 4.3.1. Morphometry and Gonadal Histological Examination

Six fish were selected from each aquarium and anesthetized. Biological indicators, including the total lengths, body lengths, and body weights, of all fish in the treatment and control groups were measured (*n* = 6). The testes of 6 fish from each group (*n* = 6, a total of 18 fish from three replicates) were cut in half and fixed with Bouin’s solution for testicular morphological analysis. The testes were fixed for 24 h, following which they were dehydrated with alcohol and cleared with xylene. Testicular tissues were then embedded in wax blocks, and continuous wax strips of 6 μm were prepared from these blocks using the Leica M2245 instrument (Leica Biosystems, Wetzlar, Germany). These were subjected to H&E staining before being observed and photographed under an RCH1-NK50I (Nikon Corporation, Tokyo, Japan) light microscope.

#### 4.3.2. Detecting Steroid Hormone Levels

According to previously published methods [70], trunks of the sampled individuals (*n* = 3 per group) were severed from the tails and immediately transferred to heparinized centrifuge tubes. Protease inhibitor (2 trypsin inhibitor units/mL) was added to the tubes, which were then centrifuged at 21,380× *g* for 15 min at 4 °C. The supernatants were carefully pipetted out and stored at −80 °C for further determination of vitellogenin (Vtg), follicle-stimulating hormone (FSH), luteinizing hormone (LH), 11-ketotestosterone (11-KT), 17β-estradiol (E2), and testosterone (T) levels, performed using commercial ELISA kits (Nanjing Jiancheng Biotechnology Co., Ltd., Nanjing, China), according to the manufacturer’s protocols.

#### 4.3.3. RNA Sample Collection and Qualification Test

We used TRIzol reagent (Invitrogen, Carlsbad, CA, USA) to extract the total RNA from each testis sample, and RNA extracted from individuals in the same aquarium was mixed to obtain a composite sample. The integrity and quantity of the RNA samples were evaluated using the Agilent 2100 Bioanalyzer (Agilent Technologies, Santa Clara, CA, USA). We obtained a total of 12 samples corresponding to the 12 aquariums, with three samples per test group, for mRNA (*n* = 3 per group, with 9 individuals in total) and miRNA (*n* = 3 per group) sequencing (Illumina HiSeq 4000, Guangzhou GeneDenovo Biotechnology Co., Ltd., Guangzhou, China).

### 4.4. RNA- and miRNA-Seq

Approximately 3 mg of total RNA (a balanced mix of 3 RNA) was used to establish libraries using the TruSeq Stranded mRNA LTSample Prep Kit (Illumina, San Diego, CA, USA), according to the manufacturer’s instructions. The cDNA libraries were then sequenced on the Illumina Hiseq 4000 platform in two lanes, and 100 bp single-end reads were generated. Total RNA extraction, RNA quantity assessment, library construction, and RNA-sequencing were performed following previously published methods and protocols [4,5]. Gene abundance was calculated and normalized to reads per kb per million reads (RPKM), using DESeq2 and EBSeq softwares. “Up_diff” or “down_diff” were classified according to the RPKM values, whereas for miRNA-seq, all clean tags were mapped to the reference transcriptome (GenBank Release 209.0, Rfam database 11.0, and miRBase database 21.0 for known miRNAs) to identify and remove rRNA, scRNA, snoRNA, snRNA, and tRNA from the miRNA sequences [10,71]. Novel miRNA candidates were identified using Mireap_v0.2 software, and the expression levels of known and novel miRNAs were calculated and normalized to transcripts per million (TPM). Principal component analysis (PCA) was performed using R package models.

### 4.5. Differentially Expressed Genes (DEGs) and DE miRNAs (DEMs)

We identified DEGs and DEMs with a fold change ≥2 and a false discovery rate (FDR) ≤0.05 from comparisons across samples or groups. These common target genes from RNA- and miRNA-seq data were further analyzed. Gene ontology (GO) enrichment and pathway-based analysis were also conducted. RNAhybrid (v2.1.2) + svm_light (v6.01), Miranda (v3.3a), and TargetScan (Version 7.0) were used to predict target gene-miRNA pairs [72].

Short time-series expression miner (STEM) was used to finally reveal the expression tendencies of DEGs, and weighted gene co-expression network analysis (WGCNA v1.47) was used to construct co-expression networks [73]. We analyzed and identified the biological function of each miRNA-mRNA pair with a negative correlation.

### 4.6. qRT-PCR Verification of Selected miRNAs and mRNAs

Based on the results of the bioinformatics analysis, miR-122-x, miR-574-x, miR-430-y, miR-217-x, lin-4-x, miR-7-y, and their target genes in the immune response/steroidogenesis pathway were selected for qRT-PCR validation (*n* = 6 per group). *Soat2* (sterol O-acyltransferase 2-like isoform X1, steroid biosynthesis and lipid metabolism), *inhbb* (inhibin beta B chain-like gene, cytokine–cytokine receptor interaction), *ihhb* (Indian hedgehog B protein-like isoform X1, hedgehog signaling pathway), *gatm* (glycine amidinotransferase, arginine, and proline metabolism), *faxdc2* (fatty acid hydroxylase domain-containing protein 2, biosynthesis of antibiotics), *ebp* (3βhydroxysteroid-Δ8,7-isomerase, biosynthesis of secondary metabolites), and *cyp1a1* (cytochrome P450 1a, retinol metabolism) were selected for qRT-PCR verification. For miRNAs, total RNAs from the testes of *G. rarus* (*n* = 3 per group) were subjected to reverse transcription using the M5 miRNA cDNA Synthesis Kit (Mei5 Biotechnology, Co., Ltd., Beijing, China) synchronously with poly (A) tail adding and complementary DNA synthesis reaction, following the manufacturer’s protocol (Mei5 Biotechnology, Co., Ltd.). qRT-PCR was conducted to determine miRNA expression using the M5 miRNA qRT-PCR Assay Kit (Mei5 Biotechnology, Co., Ltd., Beijing, Chima) and the CFX Connext^TM^ Real-TimeSystem (Bio-Rad, Hercules, CA, USA). cDNA was obtained by reverse transcription of the total RNA of the testes using the PrimeScript™ RT reagent Kit with the gDNA Eraser RR047Q (Takara Bio Inc., Dalian, China). qRT-PCR was conducted to determine mRNA expression using TB Green^®^ Premix Ex Taq™ II RR820A (Takara Bio Inc., Dalian, China) and the CFX Connext^TM^ Real-TimeSystem (Bio-Rad). The results of the miRNA and qRT-PCR analyses were compared to determine consistency and to confirm the miRNAs associated with steroid hormone synthesis and immune response in *G. rarus*. Each sample was run in triplicate, and the average threshold cycle (Ct) was used to calculate expression levels by the 2^−ΔΔCt^ method. U6snRNA and *β-actin* were used as endogenous controls (primers are listed in Appendix A) for the selected miRNAs and mRNAs, respectively. Then, correlation analysis for the miRNAs and mRNAs was performed using Pearson’s correlation coefficient.

### 4.7. Dual Luciferase Reporter Assay

The luciferase assay was performed (Ding et al., 2019 [74]) to prove the direct binding of miR-122-x and miR-574-x to the putative binding sites in the 3′-UTRs of the *faxdc2* mRNAs, miR-430-y, lin-4-x, and miR-7-y to the putative binding site in the 3′-UTR of the *inhbb* mRNA, and miR-217-x to the putative binding site in the 3′-UTR of the *ihhb* mRNA. The wild recombinant vector (pmirGLO) and mutant recombinant vector were constructed by GENERAL BIOL (GENERAL BIOL, Chuzhou, China), and the corresponding miRNA mimics were synthesized by Sangon Biotech (Sangon Biotech, Shanghai, China). Wild-type recombinant vectors contained the corresponding miRNA binding sites for *faxdc2*, *inhbb*, and *ihhb*. The expression vector could express both firefly luciferase and sea kidney luciferase. 293T cells were co-transfected with 50 nM miRNA mimics or negative controls and target genes-WT or target genes-MuT using Lipofectamine 2000 (Invitrogen, Carlsbad, CA, USA). After 48 h, relative luciferase activity was measured using the Dual-Luciferase^®^ Reporter Assay System (Promega, Madison, WI, USA), according to the manufacturer’s instructions.

### 4.8. Data Analyses

Data were analyzed using one-way ANOVA with Dunn’s post hoc test. Results were considered statistically significant for *p* < 0.05 (*) and highly statistically significant for *p* < 0.01 (**). All statistical analyses were performed using IBM SPSS Statistics 22.0 (IBM Inc., Chicago, IL, USA).

## Figures and Tables

**Figure 1 ijms-24-04239-f001:**
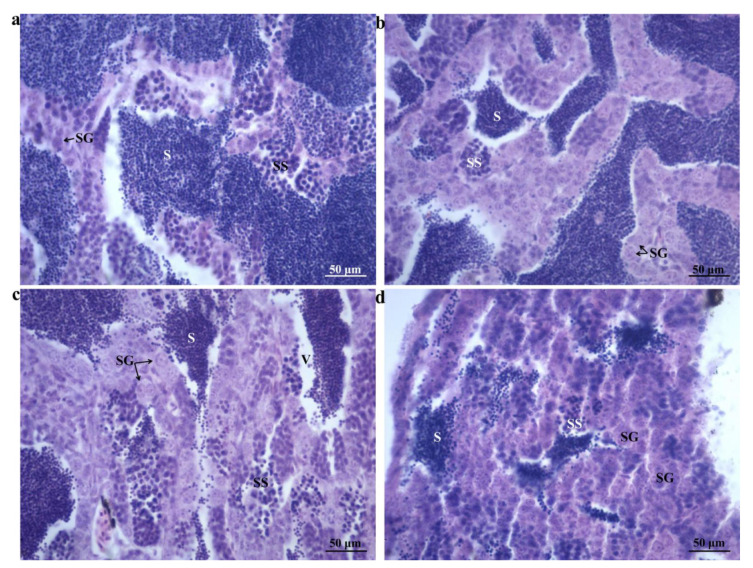
Photomicrographs of transverse testis sections of adult *G. rarus* unexposed and exposed to 17MT (*n* = 6): (**a**) 7-day control group testes, (**b**) 25 ng/L 17MT for 7-day exposure, (**c**) 50 ng/L 17MT for 7-day exposure, and (**d**) 100 ng/L 17MT for 7-day exposure. H&E staining; scale bars = 50 μm. S: mature sperm, SS: secondary spermatocyte, SG: spermatogonium, V: vacuolation.

**Figure 2 ijms-24-04239-f002:**
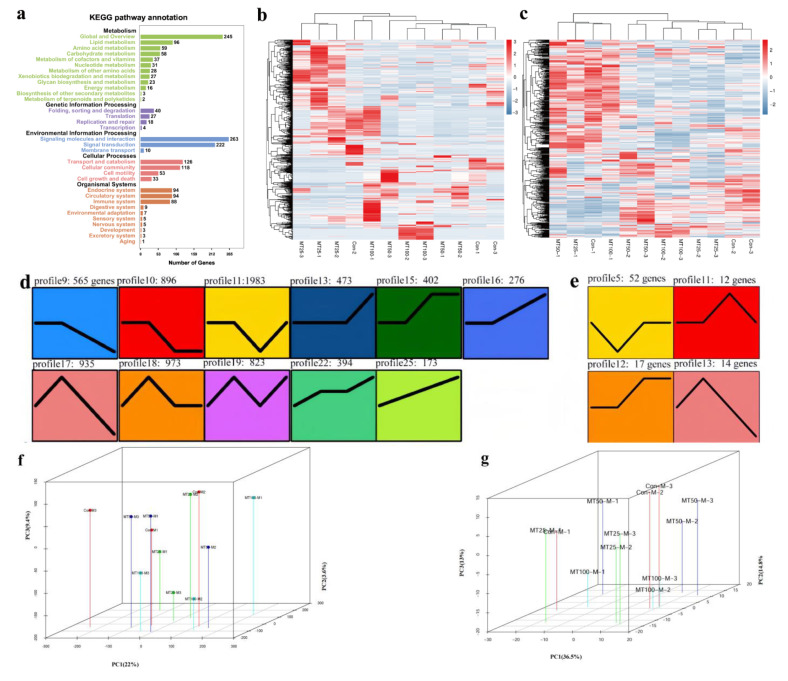
STEM analysis of RNAs (**a**,**b**,**d**,**f**) and miRNAs (**c**,**e**,**g**). (**a**) Enriched KEGG pathway annotation of RNA. Statistical summary of distribution of genes in each pathway for each trend. Rich factor refers to the ratio of the number of genes in the pathway entry in the differentially expressed genes to the total number of genes in the pathway entry in all genes. The larger the rich factor, the higher the degree of enrichment. Q-value is the *p*-value corrected by multiple hypothesis testing, and the value range is 0 to 1. The closer to zero, the more significant the enrichment is. The graph is drawn with the top 20 pathways sorted by Q-values, from small to large. (**b**) Heatmaps of RNAs (*n* = 3). (**c**) Heatmaps of miRNAs (*n* = 3). (**d**) STEM analysis of RNAs (*n* = 3). (**e**) STEM analysis of miRNAs (*n* = 3). The IDs of the trends and the numbers of genes for the trends are shown at the top of the figure; trend blocks with color (*p* < 0.05): the trends with significant enrichment and the trend blocks with similar trends have the same color. (**f**) PCA analysis of RNAs. (**g**) PCA analysis of miRNAs. The variation between samples was constrained in the PCA analysis (22.0% and 36.5% of the variance for RNAs and miRNAs alone, PC1; *p* < 0.05). In both panels, different colors correspond to samples from different 17MT concentrations. The percentage of variation explained by each axis refers to the proportion of the total data variance explained by the constrained factor.

**Figure 3 ijms-24-04239-f003:**
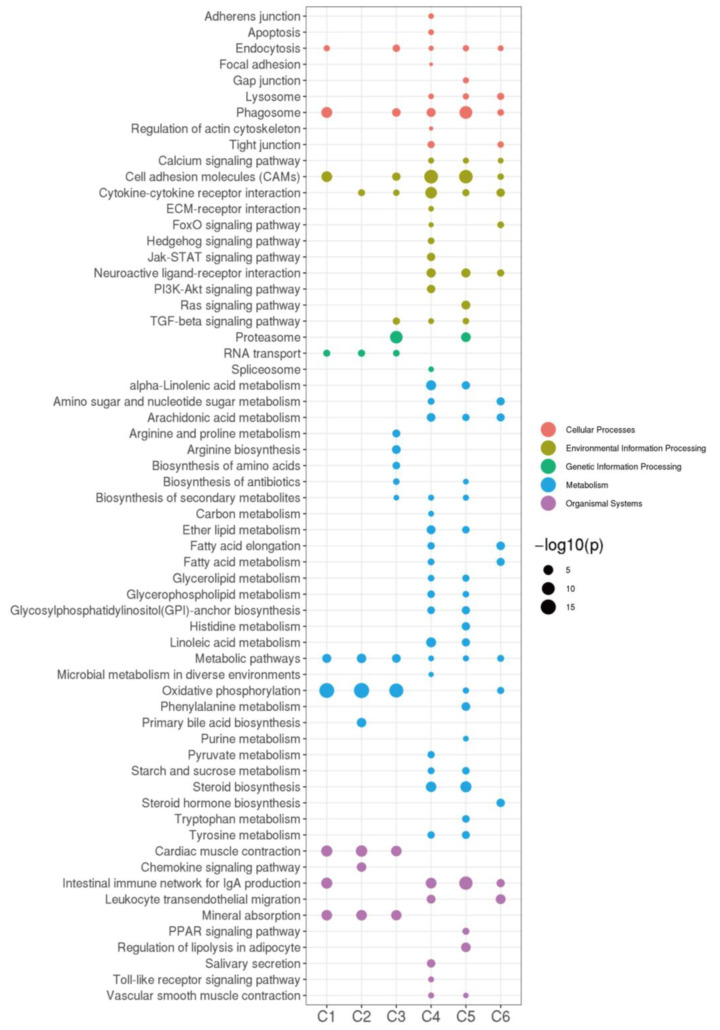
The enriched KEGG pathway annotation for miRNA-mRNA pairs as determined by WGCNA analysis using Q-values. C1, Con-M-VS-MT25-M; C2, Con-M-VS-MT50-M; C3, Con-M-VS-MT100-M; C4, MT25-M-VS-MT50-M; C5, MT25-M-VS-MT100-M; C6, MT50-M-VS-MT100-M.

**Figure 4 ijms-24-04239-f004:**
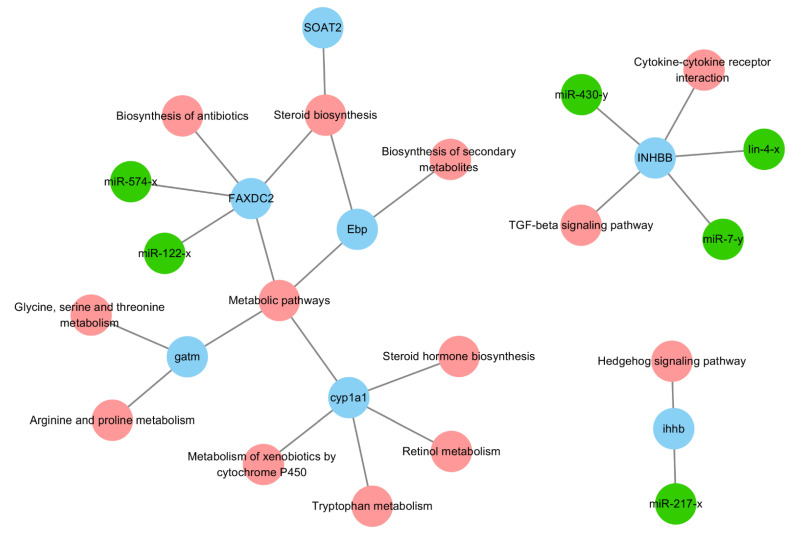
The networks of the selected miRNAs and mRNAs.

**Figure 5 ijms-24-04239-f005:**
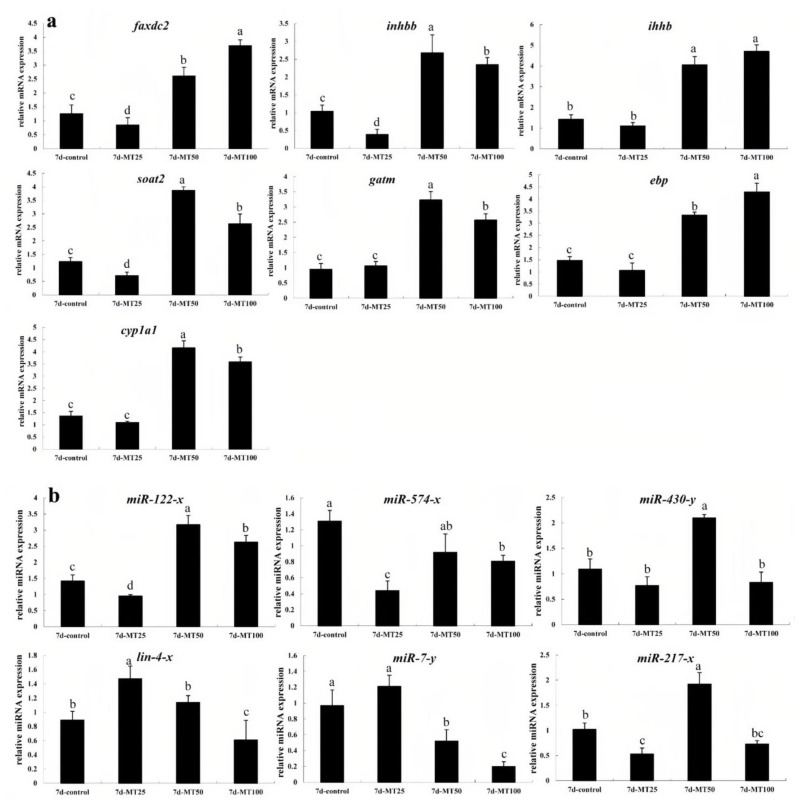
qRT-PCR verifications (*n* = 3): (**a**) testicular RNA, (**b**) testicular miRNA expression. The gene names and potential functions are as follows: *soat2* (sterol O-acyltransferase 2-like isoform X1, steroid biosynthesis and lipid metabolism), *inhbb* (inhibin beta B chain-like, cytokine–cytokine receptor interaction)*, ihhb* (indian hedgehog B protein-like isoform X1, hedgehog signaling pathway)*, gatm* (glycine amidinotransferase, arginine and proline metabolism)*, faxdc2* (fatty acid hydroxylase domain-containing protein 2, biosynthesis of antibiotics), *ebp* (3βhydroxysteroid-Δ8,7-isomerase, biosynthesis of secondary metabolites) and *cyp1a1* (cytochrome P450 1a, retinol metabolism). The miRNA names were as follows: miR-122-x, miR-574-x, miR-430-y, miR-217-x, lin-4-x, and miR-7-y. Statistically significant differences are indicated by different letters (*p* < 0.05).

**Figure 6 ijms-24-04239-f006:**
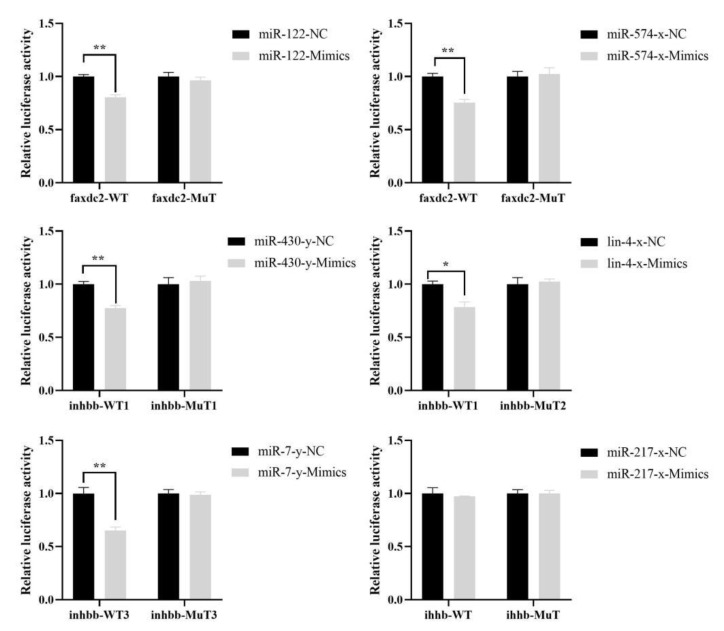
MiRNAs inversely modulated the luciferase activity of plasmids containing wt 3′-UTR of *faxdc2*, *inhbb*, and *ihhb*. Relative luciferase activity was measured and normalized by Renilla luciferase activity. * *p* < 0.05; ** *p* < 0.01.

**Table 1 ijms-24-04239-t001:** Biological indicators and the contents of hormones in the treatment and control groups.

Group	Body Length (cm)	Total Length (cm)	Total Weight	VTG (ng/mg)	FSH (mIU/mg)	11-KT (pg/mg)	E2 (pg/mg)	LH (mIU/mg)	T (pg/mg)
Control	3.64 ± 0.43	5.22 ± 0.31	1.36 ± 0.32	282.12 ± 15.05	61.49 ± 0.96	43.56 ± 2.04	5.37 ± 0.23	19.98 ± 0.72	78.31 ± 3.09
25 ng/L MT	3.52 ± 0.10	4.86 ± 0.27	1.37 ± 0.26	303.70 ± 18.13	75.91 ± 0.68 *↑	51.99 ± 0.65 **↑	7.10 ± 0.06 **↑	22.61 ± 0.68	75.83 ± 1.72
50 ng/L MT	3.44 ± 0.47	4.80 ± 0.63	1.44 ± 0.65	209.75 ± 10.14 *↓	41.65 ± 3.70 **↓	31.80 ± 0.32 **↓	4.26 ± 0.18 **↓	11.15 ± 0.91 **↓	70.20 ± 7.21
100 ng/L MT	3.38 ± 0.33	1.24 ± 0.35	3.30 ± 0.33	185.37 ± 4.21 **↓	46.53 ± 2.60 *↓	36.11 ± 1.129 **↓	4.31 ± 0.20 *↓	9.22 ± 1.23 **↓	55.20 ± 0.95 *↓

The asterisks represent statistically significant differences (* *p* < 0.05; ** *p* < 0.01). The arrows ‘↑’ or ‘↓’ indicate the significant up-regulation or down-regulation of the biological indicators and the contents of hormones from the control groups.

## Data Availability

Not applicable.

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
