# Peer review of "Comparative Analysis of miRNA-mRNA Regulation in the Testes of Gobiocypris rarus following 17α-Methyltestosterone Exposure"

_ijms, 2023, doi:10.3390/ijms24044239_

Round 1

Reviewer 1 Report

The authors present an interesting in-depth analysis of the impact of 17MT on the miRNA-RNA regulation in the testes of G. rarus. The topic is introduced well with detailed complex analyses and their results. But certain sections in the paper need to be amended to improve its readability.

Specific comments based on a section:

Materials and Methods

Section 2.3.1 – From the paragraph it is unclear how many samples were used for the morphometry and gonadal histological examination. The confusion stems from the use of n=6 in this paragraph vs n=3 indicating number of replicates being used in the other parts of the paper. Because of this confusion, this paragraph will benefit from being re-written.

The sequencer listed in section 2.3.3 Illumina HiSeq 2500 is different from the sequencer listed in section 2.4 Illumina HiSeq 2000. Please fix the discrepancy in the sequencer used.

Results

Because of the complexity of the analysis performed, there is a lot of information presented in this section with a lot of supplementary tables and figures. The information in this section needs to be clearly represented.

Section 3.1 The last sentence in this paragraph – “The results showed that with the increasing ……” is incomplete. Please complete the sentence.

What do the asterisks in Table 1 represent?

In figure 2(b) Con-2 is clustered with group MT100-1 and figure 2(c) Con-1 is clustered with group MT100-1. Can the authors explain these results?  

Section 3.4 In this section there are references to figures 3.(a) (b) (c) which don’t exist in the paper. Are these supposed to be references to Fig 2 instead? The paragraph on page 9 describes Figure (3) instead. Please add reference to Fig (3) here.

Reviewer 2 Report

In the article entitled “Comparative analysis of miRNA-mRNA regulation in the testes of Gobiocypris rarus following 17α-methyltestosterone exposure”, the authors investigated the effects of 17α-Methyltestosterone (17MT) on male Gobiocypris rarus by analizing miRNA- and RNA-seq data to determine miRNA-target gene pairs, and then they developed miRNA-mRNA interactive networks after 17MT administration. They verified that increasing amount of 17MT lead to more mature sperm (S) and fewer secondary spermatocyte (SS) and spermatogonia (SG) in the testes of control groups, as well as to less and less mature sperm (S) in the testes of male G. rarus.

They identified several both miRNAs, and mRNAs in the gonads of G. rarus, and highlighted the role of miRNA-mRNA pairs in the regulation of testicular development and immune response to disease, which will facilitate future studies on miRNA-RNA-associated regulation of teleost reproduction.

Their data might shed light on an interesting network of correlation between miRNAs, and mRNAs.

Data reported are interesting and can be taken in consideration for publication. The paper is very well written and the conclusions are exhaustively supported. However the manuscript can be considered for publicaton in after to be revised in some points, as reported here:

The authors reported an ABBREVIATIONS Section, in order to immediately follow the meaning of the acronyms reported within the text. They have to improve it by adding some terms not showed in this Section (ie. FSH, 11-KTM and so on).

In the MATERIALS and METHODS Section, sub-paragraph named “2.1. Ethics statement”, the authors didn’t reported the Shanxi Agricultural University Animal Care and Ethical Committee, China protocol authorization number. Please absolutely add it to the text.

In the MATERIALS and METHODS Section, sub-paragraph named “2.3.1. Morphometry and gonadal histological examination”, the authors reported that “Three fish were selected from each aquarium and anesthetized. Biological indicators, including the total length, body length, and body weight, of all fish in the treatment and the control groups were measured (n=6).” It is not clear the correlation between “Three fish” and “(n=6)”. Please clarify that.

In the MATERIALS and METHODS Section, sub-paragraph named “2.3.2. Detecting steroid hormone levels”, the authors reported that “…which were then centrifuged at 21,380 ×g for 15 min at -20℃”. How did they centrifuge at -20℃? Please clarify.

In the RESULTS Section, sub-paragraph named “3.1. Morphological changes”, the authors reported that “The results showed that with the increasing of secondary spermatocyte (SS) and spermatogonia (SG) in the testes of 17MT exposure groups (Fig. 1b, c and d)”. The sentence seems to be truncated? Please verify the meaning of it or remove “that with”.

In FIGURE 1, please change the colour of the text in order to evidence it on the images.

In the RESULTS Section, sub-paragraph named “3.1. Morphological changes”, the authors reported that “Significant DEGs for normalized gene expression among the differ-ent 17MT groups were identified. A total of 59 (MT25-M vs Con-M), 77 (MT50-M vs Con-M), and 66 (MT100-M vs Con-M) genes were identified as significant DEGs in the treat-ment groups (Table S3). The MT50-M and MT100-M groups presented 300 and 246 sig-nificant DEGs, respectively, in comparison to the MT25-M groups, whereas the MT100-M groups presented 135 significant DEGs when compared to the MT50-M groups. For miRNA-seq, 49 (MT25-M vs Con-M), 66 (MT50-M vs Con-M), and 49 (MT100-M vs Con-M) DEMs were identified in the treatment groups (Fig S1a). Of these, 11, 21, and 20 DEMs could be annotated to known miRNA, whereas 38, 28, and 46 DEMs were annotated to novel miRNA”. Please add a supplementary table by showing the commented raw data in order of stastitical significance (not only the top scored).

In the RESULTS Section, sub-paragraph named “3.4. STEM analysis”, the authors reported that “In RNA-seq, 565, 896, 1983, 473, 402, 276, 935, 973, 823, 394, and 173 DEGs were iden-tified in profiles 9, 10, 11, 13, 15, 16, 17, 18, 19, 22, and 25, respectively, by STEM analysis (P < 0.05, Fig. 3a)”. What did the authors meant with the term “profile”????  Please clarify that.

In the RESULTS Section, sub-paragraph named “3.4. STEM analysis”, the authors reported that “In RNA-seq, 565, 896, 1983, 473, 402, 276, 935, 973, 823, 394, and 173 DEGs were iden-tified in profiles 9, 10, 11, 13, 15, 16, 17, 18, 19, 22, and 25, respectively, by STEM analysis (P < 0.05, Fig. 3a)”. Please add a supplementary table by showing the commented raw data in order of stastitical significance.

In the RESULTS Section, sub-paragraph named “3.4. STEM analysis”, the authors reported that “Metabolic pathways (antibiotics and secondary metabolites), regulation of actin cytoskeleton, neuroactive ligand-receptor interaction, phagosomes, and cell adhesion molecules (CAMs) were enriched in almost all profiles (Fig. 3c). In miRNA-seq, 51, 12, 17, and 14 DEMs were identified in profiles 5, 11, 12, and 13, respectively (P < 0.05, Table S5, Fig.3b)”. FIGURE 3 is composed by only one image. I think the authors made a wrong list of the commented figure: Figures 3b and 3c are instead 2b and 2c; Figure 4 instead 3. Figure 2c-d-e-f-g and real Figure 4 are not commented. Please check figures number within the text and eventually correct them.

In FIGURES 5a and 5b, please add the reference on the y-axis (fold change o RQ?).

Have the authors done a correlation test, such as a Pearson correlation coefficient, between miRNAs and mRNAs whose expressions are reported in Figs. 5a and 5b, in order to verify if there is a significant inverse correlation between their expressions?

The authors commented the possible correlation between miRNAs in Fig.5b and mRNAs in Fig. 5a to support the idea of a strictly connected regulatory network. Have the authors found any evidences of direct regulations of miRNAs upon mRNAs target by seeing Luciferase assays results. I think it is very important to support their conclusions by functional assays, that lack in this manuscript.

In the DISCUSSION Section, the authors reported that “we successfully identified 6 miRNA-target pairs, which may possibly be involved in de-velopment, metabolic processes, and disease prevention. faxdc2 was the target gene regu-lated by miR-122-x and miR-574-x, whereas inhbb was the target gene regulated by miR-430-y, lin-4-x, and miR-7-y, and ihhb was the target gene regulated by miR-217-x”. I don’t see any data about the functional correlation but only data about a inverse behaviour in the expression of miRNAs and mRNAs. Please deeply improve this aspect of the manuscript.

Finally, it is not clear from where the auhtors took genomic and transcriptomic data about the Gobiocypris rarus reference genome/transcriptome. How did the identify the miRNAs? Please clearly comment these data within the manuscript.

In conclusion, this manuscript may be interesting but some aspects have to be deeply improved to support authors’idea and hypothesis.

I think the paper needs MAJOR REVISIONS in order to be considered for publication.

Round 2

Reviewer 2 Report

In the 2nd version of the article entitled “Comparative analysis of miRNA-mRNA regulation in the testes of Gobiocypris rarus following 17α-methyltestosterone exposure”, the authors responded to my comments.

With respect to my comments, they responded to almost all my questions and I appreciate very much their willingness to improve the manuscript by adding clarifying images and improve its content. Now the paper results clearer than in the first version, but there are some minor issues need to be clarified:

With regards to my comment: “The authors commented the possible correlation between miRNAs in Fig.5b and mRNAs in Fig. 5a to support the idea of a strictly connected regulatory network. Have the authors found any evidences of direct regulations of miRNAs upon mRNAs target by seeing Luciferase assays results. I think it is very important to support their conclusions by functional assays, that lack in this manuscript”, the authors “added the Dual Luciferase reporter assay. The results of the Luciferase assay showed that miR-122-x and miR-574-x target faxdc2; miR-430-y, lin-4-x, and miR-7-y target inhbb. But miR-217-x does not bind to the putative binding site in the 3’-UTR of the ihhb mRNA (Fig. 1, Figure 6 in the revised manuscript)”. I ask the authors will add a schematic representation of the miRNAs-mRNAs binding regions analyzed and report both the wt and the mutated target regions used in the reporter assays.

With regards to my comment: “Have the authors done a correlation test, such as a Pearson correlation coefficient, between miRNAs and mRNAs whose expressions are reported in Figs. 5a and 5b, in order to verify if there is a significant inverse correlation between their expressions?”, the authors replied that “The Luciferase assay was performed to prove the direct binding of miRNAs to the putative binding site in the 3’-UTR of the faxdc2, inhbb, and ihhb mRNA. Therefore, we did not add a correlation test between miRNAs and mRNAs”. Ok, that’s good. But a correlation test, such as a Paerson, is fundamental to link the two elements (miRNAs and mRNAs) in a close relationship in clinical samples like these. Please do this test and add data to the manuscript.

I think the paper would meet the journal aims and can be considered for publication after these improvements.
